# Distinct Peculiarities of *In Planta* Synthesis of Isoprenoid and Aromatic Cytokinins

**DOI:** 10.3390/biom10010086

**Published:** 2020-01-05

**Authors:** Vladimir E. Oslovsky, Ekaterina M. Savelieva, Mikhail S. Drenichev, Georgy A. Romanov, Sergey N. Mikhailov

**Affiliations:** 1Engelhardt Institute of Molecular Biology, Russian Academy of Sciences, Vavilov Str. 32, 119991 Moscow, Russia; vladimiroslovsky@gmail.com (V.E.O.); mdrenichev@mail.ru (M.S.D.); 2Timiryazev Institute of Plant Physiology, Russian Academy of Sciences, Botanicheskaya 35, 127276 Moscow, Russia; savelievaek@yandex.ru

**Keywords:** cytokinins, biosynthesis, cytokinin nucleosides, cytokinin activity, plant hormones

## Abstract

The biosynthesis of aromatic cytokinins *in planta*, unlike isoprenoid cytokinins, is still unknown. To compare the final steps of biosynthesis pathways of aromatic and isoprenoid cytokinins, we synthesized a series of nucleoside derivatives of natural cytokinins starting from acyl-protected ribofuranosyl-, 2′-deoxyribofuranosyl- and 5′-deoxyribofuranosyladenine derivatives using stereoselective alkylation with further deblocking. Their cytokinin activity was determined in two bioassays based on model plants *Arabidopsis thaliana* and *Amaranthus caudatus*. Unlike active cytokinins-bases, cytokinin nucleosides lack the hormonal activity until the ribose moiety is removed. According to our experiments, ribo-, 2′-deoxyribo- and 5′-deoxyribo-derivatives of isoprenoid cytokinin *N*^6^-isopentenyladenine turned *in planta* into active cytokinins with clear hormonal activity. As for aromatic cytokinins, both 2′-deoxyribo- and 5′-deoxyribo-derivatives did not exhibit analogous activity in *Arabidopsis*. The 5′-deoxyribo-derivatives cannot be phosphorylated enzymatically *in vivo*; therefore, they cannot be “activated” by the direct LOG-mediated cleavage, largely occurring with cytokinin ribonucleotides in plant cells. The contrasting effects exerted by deoxyribonucleosides of isoprenoid (true hormonal activity) and aromatic (almost no activity) cytokinins indicate a significant difference in the biosynthesis of these compounds.

## 1. Introduction

Cytokinins (CKs) are a group of phytohormones that play a crucial role in many processes of plant growth and development. One of the most important effects of CKs is the stimulation of plant cell division and growth. CKs promote the formation of shoots, control the root development, stimulate seed germination and the formation of pigments, activate the chloroplast formation, etc. [1,2]. Naturally occurring CKs are adenine derivatives with a hydrophobic substituent at the *N*^6^ position. CKs are divided into two groups depending on the structure of the *N*^6^ substituent: (i) aliphatic or isoprenoid, including *N*^6^-isopentenyladenine (iP) and zeatins (Figure 1A), and (ii) aromatic, including *N*^6^-benzyladenine (BA), topolins, and *N*^6^-furfuryladenine (kinetin) (Figure 1B) [1,2].

Since the discovery of kinetin in 1955 by Miller, Skoog et al. [3], a large number of various CKs as well as corresponding nucleoside and nucleotide derivatives were synthesized or isolated from different plant sources [4,5]. Among these numerous molecular forms, only CKs as nucleobases possess hormonal activity, but not their nucleoside or nucleotide derivatives [6,7]. The structural diversity of CKs and CK derivatives *in vivo* is obviously due to complex pathways of their biosynthesis.

To date, the CK biosynthesis pathways are not completely deciphered. The biosynthesis of isoprenoid CKs, in particular iP, is most studied, while the biosynthesis pathway of aromatic CKs is practically unknown [5,8]. The key step of the iP biosynthesis is commonly assumed to be isopentenyl transferase-catalyzed transfer of isopentenyl moiety from dimethylallyl diphosphate (DMAPP) to the *N*^6^ position of adenosine-5′-monophosphate (AMP), adenosine-5′-diphosphate (ADP) or adenosine-5′-triphosphate (ATP), with the formation of the respective 5′-nucleotides (iPRMP, iPRDP, iPRTP) (Figure 2). iPRDP and iPRTP are able to dephosphorylate to iPRMP. At present, two pathways of iP formation are known: (i) 5′-ribonucleotide phosphohydrolase (5′-nucleotidase) catalyzed conversion of iPRm.p. into *N*^6^-isopentenyladenosine (iPR), followed by hydrolysis of the *N*-glycosidic bond catalyzed by adenosine nucleosidase. The reverse process of ribosylation with the formation of nucleosides is catalyzed by purine nucleoside phosphorylase, and the process of reverse phosphorylation of nucleosides with the formation of nucleotides is catalyzed by adenosine kinase; (ii) iPRm.p. can be directly cleaved to iP by specific enzyme phosphoribohydrolase (LOG). Importantly, LOG specifically binds only the 5′-monophosphates of cytokinin nucleosides, but not di- or triphosphates, Am.p. or cytokinin ribosides. The reverse process of iPRm.p. formation from iP is catalyzed by adenosine phosphoribosyltransferase [9].

Notably, blocking the LOG-dependent pathway in complete knockout *log* mutants leads to severe cytokinin deficiency phenotype in *Arabidopsis* plants [10]. Therefore, although iPR is widely occurring in plants, it is considered to be mainly converted to iP via the second (i) LOG-dependent pathway (Figure 2, conversion 6) after reverse phosphorylation to 5′-monophosphate by adenosine kinase (Figure 2, conversion 3) [11].

In the present study, a new approach was proposed for the comparative analysis of the biosynthesis pathway of aromatic and isoprenoid CKs. We synthesized a series of ribo-, 2′-deoxyribo- and 5′-deoxyribonucleoside derivatives of natural iP, BA, KIN, and also synthetic cytokinin analog *N*^6^-phenylethyladenine, which possesses a high hormonal activity [7]. To investigate the possibility of biochemical conversion *in vivo* into active CKs in comparison with corresponding ribonucleosides, a series of 2′-deoxyribonucleosides was obtained. It is well known that nucleosides are converted *in vivo* to 5′-mono-, di-, and triphosphates by cellular nucleoside kinases, and the biological activity of most nucleoside analogs is associated with this mechanism [12]. To study the role of 5′-phosphorylation of nucleosides in the mechanism of CK biosynthesis, a series of 5′-deoxyribonucleoside derivatives was obtained, which cannot be phosphorylated *in vivo* to form nucleotide 5′-monophosphates and converted into active CKs via LOG-mediated cleavage (Figure 2, conversion 6).

## 2. Materials and Methods 

### 2.1. Synthesis

#### 2.1.1. General

The reagents and solvents were reagent grade (Sigma Aldrich, St. Louis, MO, USA; Merck, Darmstadt, Germany; Alfa Aesar, Haverhill, MA, USA). Column chromatography was performed on silica gel (Kieselgel 60 Merck, 0.063–0.200 mm). Thin-layer chromatography (TLC) was performed on silica-coated aluminum plates with fluorescent indicator (Merck silica gel 60F254, Darmstadt, Germany or Alugram SIL G/UV254 Macherey-Nagel, Düren, Germany with UV visualization.

^1^H and ^13^C (with complete proton decoupling) NMR spectra were recorded on Bruker AMX 400 NMR instrument at 303 K. ^1^H-NMR spectra were recorded at 400 MHz and ^13^C-NMR spectra at 100 MHz. Chemical shifts in ppm were measured relative to the residual solvent signals as internal standards (CDCl_3_, ^1^H: 7.26 ppm, ^13^C: 77.1 ppm; DMSO-d_6_, ^1^H: 2.50 ppm, ^13^C: 39.5 ppm). Spin-spin coupling constants (*J*) are given in Hz. Hydroxyl protons in ^1^H-NMR were assigned by deuterium exchange on upon addition of D_2_O into DMSO-*d*_6_ solutions of nucleosides, which led to disappearance of hydroxyl signals and simplification of ^1^H-NMR spectra. 

High-resolution mass spectra (HRMS) were registered on a Bruker Daltonics micrOTOF-Q II instrument (Bruker Daltonics, Billerica, MA, USA) using electrospray ionization (ESI). Samples were injected into the mass spectrometer chamber from the Agilent 1260 HPLC system (Santa Clara, CA, USA) equipped with an Agilent Poroshell 120 EC-C18 (3.0 × 50 mm; 2.7 µm) column (Santa Clara, CA, USA); flow rate 400 µL/min; samples were injected from the acetonitrile-water (1:1) solution and the column was eluted with a gradient of concentrations of acetonitrile (A) in water (B) in the following parameters: 0–15% A for 6.0 min, 15%–85% A for 1.5 min, 85%–0% A for 0.1 min, 0% A for 2.4 min. Retention times were as follows: **4**—3.9 min; **5**—3.9 min; **6**—4.5 min; **7**—3.7 min; **8**—4.1 min; **9**—4.1 min; **10**—4.2 min; **11**—3.8 min; **12**—4.4 min; **13**—4.3 min; **14**—4.5 min; **15**—4.0 min. 

Melting points were measured with Electrothermal Melting Point Apparatus IA6301 (Camlab Ltd., Cambridge, UK) and are uncorrected.

The following nucleosides were prepared according to the methods reported earlier: *N*^6^-benzyladenosine (**4**), *N*^6^-isopentenyladenosine (**5**), *N*^6^-furfuryladenosine (**7**) [13], *N*^6^-(2-phenylethyl)-adenosine (**6**), *N*^6^-benzyl-2′-deoxyadenosine (**8**), *N*^6^-(2-phenylethyl)-2′-deoxyadenosine (**10**), *N*^6^-benzyl-5′-deoxyadenosine (**12**), *N*^6^-isopentenyl-5′-deoxyadenosine (**13**), *N*^6^-(2-phenylethyl)-5′-deoxyadenosine (**14**) [14].

NMR and HPLC-HRMS data are presented in full in Appendix A.

#### 2.1.2. Typical Procedure for Preparation of Nucleosides by Alkylation with Alkyl Halides

A mixture of *N*^6^-acetyl-2′,3′,5′-tri-*O*-acetyladenosine (**1**) or *N*^6^-acetyl-3′,5′-di-*O*-acetyl-2′-deoxyadenosine (**2**) or *N*^6^-acetyl-2′,3′-di-*O*-acetyl-5′-deoxyadenosine (**3**) (0.5 mmol), 1,8-Diazabicyclo[5.4.0]undec-7-ene (DBU) (1 mmol), and corresponding bromide (1 mmol) in dry acetonitrile (5 mL) was kept at ambient temperature for 24 h. The reaction was monitored by TLC (silica gel, CH_2_Cl_2_-EtOH, 97:3). After 24 h the reaction mixture was concentrated in vacuo to dryness. The residue was diluted with ethyl acetate (20 mL) and washed successively with brine (2 × 20 mL), 10% aqueous sodium bicarbonate (20 mL) and water (2 × 20 mL). The organic layer was separated, dried over anhydrous sodium sulfate, filtered, and concentrated in vacuo. The residue was purified by column chromatography on silica gel. The column was washed with methylene chloride, the product was eluted with CH_2_Cl_2_-EtOH, 98:2. Purified acetyl-protected compound was dissolved in 4M n-PrNH_2_ in MeOH solution (50 mmol) and was left for 24 h, after which the mixture was concentrated in vacuo and the residue was purified by column chromatography on silica gel. The column was washed with CH_2_Cl_2_-EtOH, 95:5 and the product was eluted with CH_2_Cl_2_-EtOH, 90:10. The resulting product was dried for 24 h in a vacuum desiccator over phosphorous pentaoxide (P_2_O_5_).

*N*^6^-*benzyladenosine* (**4**). Yield for two steps was 72% as a powder. R_f_ = 0.32 (CHCl_3_-EtOH, 9:1 *v*/*v*). m.p. 164–165 °C. ^1^H-NMR (400 MHz, DMSO-*d*_6_): δ = 8.40 br s (1H, *N*^6^H), 8.37 s (1H, H-8), 8.20 br s (1H, H-2), 7.37–7.17 m (5H, Ph), 5.89 d (1H, *J*_1′2′_ = 6.1 Hz, H-1′), 5.41 d (1H, *J*_OH,2′_ = 6.2 Hz, 2-OH′), 5.34 dd (1H, *J*_OH,5′b_ = 7.1 Hz, *J*_OH,5′a_ = 4.6 Hz, 5-OH′), 5.15 d (1H, *J*_OH,3′_ = 4.7 Hz, 3-OH′), 4.72 br s (2H, *N*^6^HCH_2_), 4.61 ddd (1H, *J*_2′3′_ = 5.1 Hz, *J*_2′1′_ = 6.1 Hz, *J*_2′OH_ = 6.2 Hz, H-2′), 4.15 ddd (1H, *J*_3′4′_ = 2.8 Hz, *J*_3′2′_ = 5.1 Hz, *J*_3′OH_ = 4.7 Hz, H-3′), 3.97 ddd (1H, *J*_4′5′b_ = 3.8 Hz, *J*_4′5′a_ = 3.3 Hz, *J*_4′3′_ = 2.8 Hz, H-4′), 3.67 ddd (1H, *J*_5′a5′b_ = −12.1 Hz, *J*_5′a4′_ = 3.3 Hz, *J*_5′a,OH_ = 4.6 Hz, H-5′a), 3.56 ddd (1H, *J*_5′b5′a_ = −12.1 Hz, *J*_5′b4′_ = 3.8 Hz, *J*_5′b,OH_ = 7.1 Hz, H-5′b). ^13^C NMR (100 MHz, DMSO-d6): 154.62 (C-6), 152.39 (C-2), 148.48 (C-4), 139.96 (C-8), 128.26 (Ph), 127.16 (Ph), 126.68 (Ph), 119.78 (C-5), 88.05 (C-1′), 85.96 (C-4′), 73.58 (C-2′), 70.70 (C-3′), 61.72 (C-5′), 42.99 (NHCH_2_). HRMS: *m*/*z* [M + H]^+^ calculated C_17_H_20_N_5_O_4_^+^ 358.1510, found 358.1510.

*N*^6^-*isopentenyladenosine* (**5**). Yield for two steps was 91% as a powder. R*_f_* = 0.45 (CH_2_Cl_2_-EtOH, 4:1 *v*/*v*). m.p. 144–146 °C. ^1^H-NMR (400 MHz, DMSO-*d*_6_): δ = 8.31 s (1H, H-8), 8.19 br s (1H, H-2), 7.82 br s (1H, *N*^6^H), 5.87 d (1H, *J*_1′2′_ = 6.0 Hz, H-1′), 5.40 d (1H, *J*_OH,2′_ = 6.2 Hz, 2-OH′), 5.39 dd (1H, *J*_OH,5′b_ = 7.9 Hz, *J*_OH,5′a_ = 4.6 Hz, 5-OH′), 5.33–5.27 m (1H, CCH = CMe_2_), 5.15 d (1H, *J*_OH,3′_ = 4.8 Hz, 3-OH′), 4.60 ddd (1H, *J*_2′3′_ = 5.3 Hz, *J*_2′1′_ = 6.0 Hz, *J*_2′OH_ = 6.2 Hz, H-2′), 4.14 ddd (1H, *J*_3′4′_ = 2.8 Hz, *J*_3′2′_ = 5.3 Hz, *J*_3′OH_ = 4.8 Hz, H-3′), 4.08 br s (2H, *N*^6^HCH_2_), 3.96 ddd (1H, *J*_4′5′b_ = 3.0 Hz, *J*_4′5′a_ = 3.6 Hz, *J*_4′3′_ = 2.8 Hz, H-4′), 3.67 ddd (1H, *J*_5′a5′b_ = −12.1 Hz, *J*_5′a4′_ = 3.6 Hz, *J*_5′a,OH_ = 4.6 Hz, H-5′a), 3.55 ddd (1H, *J*_5′b5′a_ = −12.1 Hz, *J*_5′b4′_ = 3.0 Hz, *J*_5′b,OH_ = 7.9 Hz, H-5′b). ^13^C NMR (100 MHz, DMSO-d6): 154.93 (C-6), 153.14 (C-2), 148.63 (C-4), 140.49 (C-8), 135.41 (=CMe_2_), 121.73 (CH=), 120.14 (C-5), 88.79 (C-1′), 86.57 (C-4′), 74.22 (C-2′), 71.20 (C-3′), 62.22 (C-5′), 38.57 (NHCH_2_), 26.02 (Me), 18.49 (Me). HRMS: *m*/*z* [M + H]^+^ calculated C_15_H_22_N_5_O_4_^+^ 336.1666, found 336.1666.

*N*^6^-*benzyl-2′-deoxyadenosine* (**8**). Yield for two steps was 55% as a foam. R*_f_* 0.08 (CH_2_Cl_2_-EtOH, 97:3). ^1^H NMR (400 MHz, DMSO-*d*_6_): δ = 8.41 (br s, 1H, *N*^6^H), 8.35 (s, 1H, H2), 8.19 (s, 1H, H8), 7.37–7.16 (m, 5H, Ph), 6.35 (dd, 1H, *J*_1′,2′b_ = 5.9 Hz, *J*_1′,2′a_ = 7.8 Hz, H1′), 5.30 (d, 1H, *J*_OH,3′_ = 3.9 Hz, 3′OH), 5.20 (dd, 1H, *J*_OH, 5′a_ = 5.2 Hz, *J*_OH,5′b_ = 6.4 Hz, 5′OH), 4.70 (br s, 2H, NHCH_2_), 4.40 (dddd, 1H, *J*_3′4′_ = 2.5 Hz, *J*_3′2′a_ = 5.7 Hz, *J*_3′2′b_ = 2.3 Hz, *J*_3′OH_ = 3.9 Hz, H3′), 3.88 (ddd, 1H, *J*_4′,3′_ = 2.3 Hz, *J*_4′,5′a_ = 3.9 Hz, *J*_4′,5′b_ = 6.2 Hz, H4′), 3.62 (ddd, 1H, *J*_5′a,4′_ = 3.9 Hz, *J*_5′a,5′b_ = −12.0 Hz, *J*_5′a,OH_ = 5.2 Hz, H5′a), 3.51 (ddd, 1H, *J*_5′b,4′_ = 6.2 Hz, *J*_5′b,5′a_ = −12.0 Hz, *J*_5′b,OH_ = 6.4 Hz, H5′b), 2.74 (ddd, 1H, *J*_2′a,1′_ = 7.8 Hz, *J*_2′a,3′_ = 5.7 Hz, *J*_2′a,2′b_ = −13.1 Hz, H2′a), 2.25 (ddd, 1H, *J*_2′b,1′_ = 5.8 Hz, *J*_2′b,3′_ = 2.3 Hz, *J*_2′b,2′a_ = −13.1 Hz, H2′b). ^13^C NMR (400 MHz, DMSO-*d*_6_): δ = 154.90 (C6), 152.86 (C2), 148.61 (C4), 140.10 (C8), 128.82 (Ph), 127.58 (Ph), 127.32 (Ph), 119.55 (C5), 88.28 (C1′), 84.62 (C4′), 71.34 (C3′), 62.20 (C5′), 43.38 (NHCH_2_), 39.49 (C2′ overlapping with DMSO). HRMS: *m*/*z* [M + H]^+^ calculated C_17_H_20_N_5_O_3_^+^ 342.1561, found 342.1562; *m*/*z* [M–deoxyribosyl]^+^ calculated C_12_H_12_N_5_^+^ 226.1087, found 226.1083.

*N*^6^-*isopentenyl-2′-deoxyadenosine* (**9**). Yield for two steps was 43% as a foam. R*_f_* 0.33 (CH_2_Cl_2_-EtOH, 95:5). ^1^H NMR (400 MHz, DMSO-*d*_6_): δ = 8.30 s (1H, H-2), 8.18 br s (1H, H-8), 7.79 br s (1H, *N*^6^H), 6.34 dd (1H, *J*_1′2′b_ = 6.0 Hz, *J*_1′2′a_ = 7.5 Hz, H-1′), 5.30 dd (1H, *J*_CHCH3_ = 1.2 Hz, *J*_CHCH2_ = 6.5 Hz, CH = CMe_2_), 5.27 d (1H, *J*_OH,3′_ = 4.0 Hz, 3-OH′), 5.18 dd (1H, *J*_OH,5′b_ = 6.4 Hz, *J*_OH,5′a_ = 4.8 Hz, 5-OH′), 4.40 dddd (1H, *J*_3′4′_ = 2.7 Hz, *J*_3′2′a_ = 5.9 Hz, *J*_3′2′b_ = 2.9 Hz, *J*_3′OH_ = 4.0 Hz, H-3′), 4.07 br s (2H, *N*^6^HCH_2_), 3.88 ddd (1H, *J*_4′5′b_ = 4.2 Hz, *J*_4′5′a_ = 4.4 Hz, *J*_4′3′_ = 2.7 Hz, H-4′), 3.62 ddd (1H, *J*_5′a5′b_ = −12.0 Hz, *J*_5′a4′_ = 4.5 Hz, *J*_5′a-OH_ = 4.8 Hz, H-5′a), 3.52 ddd (1H, *J*_5′b5′a_ = −12.0 Hz, *J*_5′b4′_ = 4.2 Hz, *J*_5′b,OH_ = 6.3 Hz, H-5′b), 2.71 (ddd, 1H, *J*_2′a,1′_ = 7.5 Hz, *J*_2′a,3′_ = 5.9 Hz, *J*_2′a,2′b_ = −13.0 Hz, H2′a), 2.25 (ddd, 1H, *J*_2′b,1′_ = 6.0 Hz, *J*_2′b,3′_ = 2.8 Hz, *J*_2′b,2′a_ = −13.0 Hz, H2′b), 1.69 (s, 3H, CH_3_-*cis*), 1.66 (s, 3H, CH_3_-*trans*). ^13^C NMR (100 MHz, DMSO-*d*_6_): δ = 154.32 (C6), 152.26 (C2), 148.09 (C4), 139.21 (C8), 133.14 (=CMe_2_), 122.07 (CH=), 119.57 (C5), 87.96 (C1′), 83.92 (C4′), 70.93 (C3′), 61.86 (C5′), 39.49 (C2′, overlapping with DMSO), 37.67 (NHCH_2_), 25.32 (CH_3_ isopentenyl), 17.77 (CH_3_ isopentenyl). HRMS: *m*/*z* [M + H]^+^ calculated C_15_H_22_N_5_O_3_^+^ 320.1717, found 320.1720; *m*/*z* [M–deoxyribosyl]^+^ calculated C_12_H_12_N_5_^+^ 204.1244, found 204.1236.

*N*^6^-*(2-phenylethyl)-2′-deoxyadenosine* (**10**). Yield for two steps was 58% as a foam. R*_f_* 0.06 (CH_2_Cl_2_-EtOH, 97:3). ^1^H NMR (400 MHz, DMSO-*d*_6_): δ = 8.32 (s, 1H, H2), 8.22 (s, 1H, H8), 7.83 (br s, 1H, *N*^6^H), 7.35–7.15 (m, 5H, Ph), 6.35 (dd, 1H, *J*_1′,2′b_ = 6.2 Hz, *J*_1′,2′a_ = 7.3 Hz, H1′), 5.27 (d, 1H, *J*_OH,3′_ = 3.9 Hz, 3′OH), 5.18 (dd, 1H, *J*_OH,5′b_ = 6.4 Hz, *J*_OH,5′a_ = 5.1 Hz, 5′OH), 4.44–4.36 (m, 1H, H3′), 3.88 (ddd, 1H, *J*_4′,3′_ = 6.6 Hz, *J*_4′,5′a_ = 4.3 Hz, *J*_4′,5′b_ = 4.1 Hz, H4′), 3.82–3.60 (m, 2H, *N*^6^HCH_2_), 3.62 (ddd, 1H, *J*_5′a,4′_ = 4.3 Hz, *J*_5′a,5′b_ = −12.0 Hz, *J*_OH,5′a_ = 5.1 Hz, H5′a), 3.52 (ddd, 1H, *J*_5′b,4′_ = 4.1 Hz, *J*_5′b,5′a_ = −12.0 Hz, *J*_OH,5′b_ = 6.4 Hz, H5′b), 2.92 (dd, 2H, *J*_CH2-CH2_ = 7.8 Hz, *J*
_CH2-CH2_ = 7.0 Hz, CH_2_Ph), 2.72 (ddd, 1H, *J*_2′a,1′_ = 7.3 Hz, *J*_2′a,3′_ = 5.6 Hz, *J*_2′a,2′b_ = −13.1 Hz, H2′a), 2.72 (ddd, 1H, *J*_2′b,1′_ = 6.1 Hz, *J*_2′b,3′_ = 2.9 Hz, *J*_2′b,2′a_ = −13.1 Hz, H2′a). ^13^C NMR (100 MHz, CD_3_OD): δ = 156.21 (C6), 153.51 (C2), 148.86 (C4), 140.93 (C8), 140.47 (Ph), 129.88 (Ph), 129.46 (Ph), 127.31 (Ph), 121.30 (C5), 89.91 (C1′), 87.15 (C4′), 73.07 (C3′), 63.67 (C5′), 43.19 (NHCH_2_), 41.58 (C2′), 36.68 (CH_2_Ph). HRMS: *m*/*z* [M + H]^+^ calculated C_18_H_22_N_5_O_3_^+^ 356.1717, found 356.1724; *m*/*z* [M–deoxyribosyl]^+^ calculated C_13_H_14_N_5_^+^ 240.1244, found 240.1241.

*N*^6^-*benzyl-5′-deoxyadenosine* (**12**). Yield for two steps was 77% as a powder. R*_f_* 0.32 (CH_2_Cl_2_-EtOH, 95:5). m.p. 198–201 °C. ^1^H NMR (400 MHz, DMSO-*d*_6_): δ = 8.29–8.37 (m, 2H, H-2, *N*^6^H), 8.21 (s, 1H, H-8), 7.16–7.37 (m, 5H, Ph), 5.85 (dd, 1H, *J*_1′-2′_ = 4.77 Hz, *J*_1′-2′OH_ = 2.54 Hz, H1′), 5.38 (d, 1H, *J*_3′-OH_ = 5.25 Hz, 3′OH), 5.10 (d, 1H, *J*_2′-OH_ = 4.61 Hz, 2′OH), 4.60–4.83 (m, 3H, H2′, NHCH_2_), 3.93–4.10 (m, 2H, H3′, H4′), 1.30 (dd, 3H, *J*_CH3-4′_= 4.77 Hz, *J*_CH3-3′OH_ = 1.4 Hz, CH_3_). ^13^C NMR (100 MHz, DMSO-*d*_6_): δ = 154.47 (C6), 152.54 (C2), 148.79 (C4), 140.04 (Ph), 139.82 (C8), 128.14 (Ph), 127.08 (Ph), 126.54 (Ph), 119.51 (C5), 87.92 (C1′), 79.71 (C4′), 74.59 (C2′), 73.05 (C3′), 42.89 (CH_2_Ph), 16.89 (CH_3_). HRMS: *m*/*z* [M + H]^+^ calculated C_17_H_20_N_5_O_3_^+^ 342.1561, found 342.1566; *m*/*z* [M–deoxyribosyl]^+^ calculated C_12_H_12_N_5_^+^ 226.1087, found 226.1086.

*N*^6^-*isopentenyl-5′-deoxyadenosine* (**13**). Yield for two steps was 47% as a powder. R*_f_* 0.31 (CH_2_Cl_2_-EtOH, 95:5). m.p. 117-120 °C. ^1^H NMR (400 MHz, DMSO-*d*_6_): δ = 8.29 (s, 1H, H2), 8.21 (s, 1H, H8), 7.77 (br s, 1H, *N*^6^H), 5.84 (d, 1H, *J*_1′-2′_ = 4.8 Hz, H1′), 5.38 (d, 1H, *J*_2′-OH_ = 5.6 Hz, 2′OH), 5.30 (tq, 1H, *J*_CH-CH2_ = 6.6 Hz, *J*_CH-CH3_ = 1.3 Hz, CH=), 5.10 (d, 1H, *J*_3′-OH_ = 5.3 Hz, 3′OH), 4.65 (ddd, 1H, *J*_2′-3′_=4.6 Hz, *J*_2′-1′_ = 4.8 Hz, *J*_2′-OH_ = 5.6 Hz, H2′), 4.08 (br s, 2H, *N*^6^HCH_2_), 3.93–4.01 (m, 2H, H3′, H4′), 1.70 (d, 3H, *J*_CH3-CH_ = 0.5 Hz, CH_3_-*cis*), 1.67 (d, 3H, *J*_CH3-CH_ = 0.8 Hz, CH_3_-*trans*), 1.30 (d, 3H, *J*_CH3-CH_ = 6.2 Hz, CH_3_). ^13^C NMR (100 MHz, DMSO-*d*_6_): δ = 154.30 (C6), 152.53 (C2), 148.43 (C4), 139.53 (C8), 133.11 (CMe_2_), 122.13 (CH=), 119.52 (C5), 87.84 (C1′), 79.63 (C4′), 74.57 (C2′), 73.05 (C3′), 37.67 (NHCH_2_), 25.33 (CH_3_), 18.86 (CH_3_ isopentenyl), 17.79 (CH_3_ isopentenyl). HRMS: *m*/*z* [M + H]^+^ calculated C_15_H_22_N_5_O_3_^+^ 320.1717, found 320.1723; *m*/*z* [M–deoxyribosyl]^+^ calculated C_12_H_12_N_5_^+^ 204.1244, found 204.1240.

#### 2.1.3. Typical Procedure for Preparation of Nucleosides by Mitsunobu Reaction with Alcohols

To the solution of *N*^6^-acetyl-2′,3′,5′-tri-*O*-acetyladenosine (**1**) or *N*^6^-acetyl-3′,5′-di-*O*-acetyl-2′-deoxyadenosine (**2**) or *N*^6^-acetyl-2′,3′-di-*O*-acetyl-5′-deoxyadenosine (**3**) (1 mmol) with triphenylphosphine (Ph_3_P) (2 mmol) and corresponding alcohol (2 mmol) in tetrahydrofuran (THF) (5 mL) diethyl azodicarboxylate (DEAD) (2 mmol) was added in one portion and the solution was kept at r.t. for 48 h. The reaction was monitored by TLC (silica gel, CH_2_Cl_2_-EtOH, 97:3). After 48 h the reaction mixture was concentrated in vacuo and the residue was dissolved in CH_2_Cl_2_ and washed with brine (3 × 20 mL). The organic layer was separated, dried over anhydrous sodium sulfate, filtered, and concentrated in vacuo. The residue was purified by column chromatography (silica gel, CH_2_Cl_2_-EtOH, 97:3). Partially purified compound was dissolved in 4M n-PrNH_2_ in MeOH solution (50 mmol) and left at r.t. for 24 h, after which the mixture was concentrated in vacuo and the residue was purified by column chromatography on silica gel. The column was washed with CH_2_Cl_2_-EtOH, 95:5, the product was eluted with CH_2_Cl_2_-EtOH, 90:10. The resulting product was dried for 24 hrs in a vacuum desiccator over phosphorous pentaoxide (P_2_O_5_).

*N*^6^-*(2-phenylethyl)-adenosine* (**6**). Yield for two steps was 65% as a powder. R*_f_* 0.28 (CH_2_Cl_2_-EtOH, 9:1 *v*/*v*). m.p. 169–171 °C. ^1^H NMR (400 MHz, DMSO-*d*_6_): δ = 8.34 (s, 1H, H8), 8.23 (br s, 1H, H2), 7.88 (br s, 1H, *N*^6^H), 7.16–7.33 (m, 5H, Ph), 5.89 (d, 1H, *J*_1′,2′_ = 6.1 Hz, H1′), 5.41 (d, 1H, *J*_OH-2′_ = 6.1 Hz, 2′OH), 5.36 (dd, 1H, *J*_OH-5′b_ = 7.0 Hz, *J*_OH-5′a_ = 4.6 Hz, 5′OH), 5.15 (d, 1H, *J*_OH-3′_ = 4.6 Hz, 3′OH), 4.60 (dd, 1H, *J*_2′,3′_ = 5.9 Hz, *J*_2′,1′_ = 6.1 Hz, H2′),4.15 (ddd, 1H, *J*_3′,4′_ = 3.2 Hz, *J*_3′,2′_ = 5.9 Hz, *J*_3′,OH_ = 4.6 Hz, H3′), 3.97 (ddd, 1H, *J*_4′,5′b_ = 3.0, *J*_4′,5′a_ = 3.4, *J*_4′,3′_ = 3.2, H4′), 3.63–3.81 (m, 3H, *N*^6^HCH_2_, H5′a), 3.55 (ddd, 1H, *J*_5′b,5′a_ = −12.0 Hz, *J*_5′b,4′_ = 3.4 Hz, *J*_5′b,OH_ = 7.0 Hz, H5′b), 2.93 (t, 2H, *J*_CH2-CH2_ = 7.5 Hz, CH_2_Ph). ^13^C NMR (100 MHz, DMSO-*d*_6_): δ = 154.61 (C6), 152.35 (C2), 148.27 (C4), 139.70 (C8), 139.48 (Ph), 128.63 (Ph), 128.27 (Ph), 126.00 (Ph), 119.74 (C5), 87.94 (C1′), 85.88 (C4′), 73.49 (C2′), 70.63 (C3′), 61.66 (C5′), 41.23 (NCH_2_), 34.39 (CH_2_Ph). HRMS: *m*/*z* [M + H]^+^ calculated C_18_H_22_N_5_O_4_^+^ 372.1666, found 372.1666.

*N*^6^-*furfuryladenosine* (**7**). Yield for two steps was 64% as a powder. R*_f_* = 0.33 (CH_2_Cl_2_-EtOH, 90:10). m.p. 150–155 °C. ^1^H NMR (400 MHz, DMSO-d6): δ = 8.36 s (1H, H-8), 8.23 br s (1H, *N*^6^H), 8.23 s (1H, H-2), 7.52 dd (1H, ^3^*J*=3.1 Hz, ^4^*J*=1.9 Hz, H5_Fur_), 6.35 dd (1H, ^3^*J*=3.1 Hz, ^3^*J*=1.9 Hz, H4_Fur_), 6.23 dd (1H, ^3^*J* = 3.1 Hz, ^4^*J*=0.7 Hz, H3_Fur_), 5.89 d (1H, *J*_1′2′_=6.2 Hz, H-1′), 5.41 d (1H, *J*_OH,2′_ = 6.2 Hz, 2′-OH), 5.34 dd (1H, *J*_OH,5′b_ = 7.0 Hz, *J*_OH,5′a_ = 4.6 Hz, 5′-OH), 5.15 d (1H, *J*_OH,3′_ = 4.8 Hz, 3′-OH), 4.71 br s (2H, *N*^6^HCH_2_), 4.61 ddd (1H, *J*_2′3′_ = 4.9 Hz, *J*_2′1′_ = 6.2 Hz, *J*_2′OH_ = 6.2 Hz, H-2′), 4.15 ddd (1H, *J*_3′4′_ = 3.0 Hz, *J*_3′2′_ = 4.9 Hz, *J*_3′OH_ = 4.8 Hz, H-3′), 3.97 ddd (1H, *J*_4′5′b_ = 3.9 Hz, *J*_4′5′a_ = 3.9 Hz, *J*_4′3′_ = 3.0 Hz, H-4′), 3.67 ddd (1H, *J*_5′a5′b_ = −12.0 Hz, *J*_5′a4′_ = 3.9 Hz, *J*_5′a,OH_ = 4.6 Hz, H-5′a), 3.56 ddd (1H, *J*_5′b5′a_ = −12.0 Hz, *J*_5′b4′_ = 3.9 Hz, *J*_5′b,OH_ = 7.0 Hz, H-5′b). ^13^C NMR (100 MHz, DMSO-d6): δ = 154.41 (C-6), 152.88 (Fur), 152.30 (C-2), 148.68 (C-4), 141.87 (Fur), 140.03 (C-8), 119.85 (C-5), 110.50 (Fur), 106.73 (Fur), 88.01 (C-1′), 85.93 (C-4′), 73.59 (C-2′), 70.66 (C-3′), 61.69 (C-5′), 36.62 (NHCH_2_). HRMS: *m*/*z* [M + H]^‒^ calculated C_15_H_18_N_5_O_5_^+^ 348.1302, found 348.1307.

*N*^6^-*furfuryl-2′-deoxyadenosine* (**11**). Yield for two steps was 58% as a foam. R*_f_* 0.52 (CH_2_Cl_2_-EtOH, 95:5). ^1^H NMR (400 MHz, DMSO-*d*_6_): δ = 8.36 s (1H, H-2), 8.23 br s (2H, H-8, *N*^6^H), 7.53 dd (1H, *J*_H2-H3_ = 1.8 Hz, *J*_H2-H4_ = 0.9 Hz, H2-furan), 6.39-6.33 m (2H, H-1′, H3-furan), 6.23 dd (1H, *J*_H4-H3_ = 3.2 Hz, *J*_H4-H2_ = 0.9 Hz, H4-furan), 5.29 d (1H, *J*_OH-3′_ = 3.9 Hz, 3-OH′), 5.16 dd (1H, *J*_OH-5′b_ = 6.2 Hz, *J*_OH-5′a_ = 4.4 Hz, 5-OH′), 4.72 br s (2H, *N*^6^HCH_2_), 4.41 dddd (1H, *J*_3′-4′_ = 2.7 Hz, *J*_3′-2′a_ = 5.8 Hz, *J*_3′-2′b_ = 2.8 Hz, *J*_3′-OH_ = 3.9 Hz, H-3′), 3.89 ddd (1H, *J*_4′-5′b_ = 6.9 Hz, *J*_4′-5′a_ = 4.6 Hz, *J*_4′-3′_ = 2.7 Hz, H-4′), 3.62 ddd (1H, *J*_5′a-5′b_ = −11.7 Hz, *J*_5′a-4′_ = 4.6 Hz, *J*_5′a-OH_ = 6.2 Hz, H-5′a), 3.52 ddd (1H, *J*_5′b5′a_ = −11.7 Hz, *J*_5′b4′_ = 6.9 Hz, *J*_5′b,OH_ = 4.4 Hz, H-5′b), 2.73 ddd (1H, *J*_2′a-1′_ = 7.8 Hz, *J*_2′a-3′_ = 5.8 Hz, *J*_2′a-2′b_ = −13.2 Hz, H2′a), 2.25 ddd (1H, *J*_2′b-1′_ = 6.1 Hz, *J*_2′b-3′_ = 2.8 Hz, *J*_2′b-2′a_ = −13.2 Hz, H2′b). ^13^C NMR (100 MHz, CDCl_3_): δ = 154.90 (C-6), 152.59 (C-2), 151.48 (C-4), 142.45 (C-8), 139.79 (Fur), 121.50 (C-5), 110.60 (Fur), 107.81 (Fur), 89.83 (C-1′), 87.86 (C-4′), 73.43 (C-3′), 63.56 (C-5′), 40.98 (C-2′), 37.72 (NHCH_2_). HRMS: *m*/*z* [M + H]^+^ calculated C_15_H_18_N_5_O_4_^+^ 332.1353, found 332.1350.

*N*^6^-*(2-phenylethyl)-5′-deoxyadenosine* (**14**). Yield for two steps was 54% as a powder; R*_f_* 0.30 (CH_2_Cl_2_-EtOH, 95:5). m.p. 161–163 °C. ^1^H NMR (DMSO-*d*_6_): δ = 8.30 (s, 1H, H2), 8.24 (s, 1H, H8), 7.79 (br s, 1H, *N*^6^H), 7.15–7.40 (m, 5H, Ph), 5.85 (d, 1H, *J*_1′-2′_ = 4.9 Hz, H1′), 5.38 (d, 1H, *J*_3′-OH_ = 5.57 Hz, 3′OH), 5.10 (d, 1H, *J*_2′-OH_ = 5.1 Hz, 2′OH), 4.66 (ddd, 1H, *J*_2′-1′_ = 4.9 Hz, *J*_2′-3′_ = 4.5 Hz, *J*_2′-OH′_ = 5 Hz, H2′), 3.92–4.02 (m, 2H, H3′, H4′), 3.72 (br s, 2H, NHCH_2_), 2.93 (t, 2H, *J*_CH2-CH2_ = 7 Hz, CH_2_Ph), 1.31 (d, 3H, *J*_CH3-4′_ = 6.04 Hz, CH_3_). ^13^C NMR (DMSO-*d*_6_): δ = 154.50 (C6), 152.58 (C2), 148.61 (C4), 139.62 (C8, Ph), 128.62 (Ph), 128.25 (Ph), 125.98 (Ph), 119.54 (C5), 87.85 (C1′), 79.67 (C4′), 74.59 (C2′), 73.07 (C3′), 41.25 (CH_2_Ph), 35.01 (NHCH_2_), 18.88 (CH_3_). HRMS: *m*/*z* [M + H]^+^ calculated C_18_H_22_N_5_O_3_^+^ 356.1717, found 356.1727; *m*/*z* [M–deoxyribosyl]^+^ calculated C_13_H_14_N_5_^+^ 240.1244, found 240.1243.

*N*^6^-*furfuryl-5′-deoxyadenosine* (**15**). Yield for two steps was 73% as a powder; R*_f_* 0.24 (CH_2_Cl_2_-EtOH, 97:3). m.p. 185–188 °C. ^1^H NMR (DMSO-*d*_6_): δ = 8.34 s (1H, H-2), 8.25 s (1H, H8), 8.22 br s (1H, *N*^6^H), 7.53 dd (1H, *J*_H2-H3_ = 1.7 Hz, *J*_H2-H4_ = 0.8 Hz, H2-furan), 6.36 dd (1H, *J*_H3-H2_ = 1.7 Hz, *J*_H3-H4_ = 3.1 Hz, H3-furan), 6.23 dd (1H, *J*_H4-H3_ = 3.1 Hz, *J*_H4-H2_ = 0.8 Hz, H4-furan), 5.86 d (*J*_1′-2′_ = 4.9 Hz, H-1′), 5.41 d (1H, *J*_OH-3′_ = 5.7 Hz, 3-OH′), 5.14 d (1H, *J*_OH-2′_ = 5.2 Hz, 2-OH′), 4.70 br s (2H, *N*^6^HCH_2_), 4.67 ddd (1H, *J*_2′-1′_ = 4.9 Hz, *J*_2′-3′_ = 4.5 Hz, *J*_2′-OH_ = 5.2 Hz, H-2′), 4.05-3.90 m (H-3′, H-4′), 1.30 d (*J*_CH3-CH_ = 6.1 Hz, CH_3_). ^13^C NMR (100 MHz, DMSO-d6): δ = 154.27 (C-6), 152.93 (C-4), 152.44 (C-2), 141.77 (C-8), 139.96 (Fur), 119.60 (C-5), 110.40 (Fur), 106.59 (Fur), 87.89 (C-1′), 79.72 (C-4′), 74.59 (C-2′), 73.05 (C-3′), 36.54 (NHCH_2_), 18.90 (CH_3_). HRMS: *m*/*z* [M + H]^+^ calculated C_15_H_18_N_5_O_4_^+^ 332.1353, found 332.1355.

### 2.2. Cytokinin Activity Assays

Investigated compounds were dissolved in 100% DMSO to a concentration of 0.1 M. Then these solutions were diluted with distilled water to concentrations of 1–10 μM. Thus, the content of DMSO in 1 μM ligand solutions was only 0.001%.

CK activities were measured in two bioassays based on seedlings of *Arabidopsis thaliana* or *Amaranthus caudatus*. In the first assay, we used double mutants of *Arabidopsis*, in which only one of the three CK receptors (AHK2, AHK3, or CRE1/AHK4) was kept active, as well as wild type (WT) *Arabidopsis* seedlings with all three functioning receptors [15]. All *Arabidopsis* plants used, including WT control, carried the reporter *GUS* gene under control of CK-dependent promoter *P_ARR5_* [16]. 4- to 5-day-old *Arabidopsis* seedlings were incubated for 16 h in aqueous solutions of tested compounds [17]. The level of CK activity of each compound was determined through the level of GUS activity, since it reflects the intensity of *P_ARR5_:GUS* expression [18]. Each sample contained 10 aligned seedlings; experiments were performed in two biological replicates.

In the case of *Amaranthus* bioassay, 3- to 4-day-old etiolated seedlings with removed roots were incubated for 16 h in solutions of tested compounds in the dark. Each sample contained 10 aligned seedlings; all samples were in triplicate. The CK activity of compounds corresponds to the level of the pigment amaranthin accumulation in cotyledons, which was determined with spectrophotometer [19,20].

The experiments were carried out in 2–3 biological replications. In all our experiments, probes with BA were included as positive control and its activity was set as 100%. CK activity of all tested compounds was evaluated as percentage relative to the BA activity (after background subtraction) at the same concentration in the same experiment.

## 3. Results and Discussion

### 3.1. Chemistry

To synthesize desired nucleosides, we used our mild and efficient approach based on regioselective *N*^6^-alkylation of acetyl-protected adenosine, 2′-deoxyadenosine and 5′-deoxyadenosine derivatives **1-3** either with alcohols under Mitsunobu reaction conditions (Scheme 1, conditions (i)) or with alkyl halides in the presence of the base (Scheme 1, conditions (ii)) [13,14]. The main advantage of these approaches is the possibility to use both alkyl halides and alcohols for *N*^6^-modification. The following deacetylation with n-PrNH_2_ in MeOH (Scheme 1, conditions (iii)) yielded the final nucleosides with high overall yields. The initial triacetyl-5′-deoxyadenosine **3** was synthesized by radical reduction of corresponding 5′-chloro-5′-deoxyadenosine derivative in the presence of Bu_3_SnH according to the literature [21].

The structure of the compounds was confirmed by ^1^H and ^13^C NMR spectroscopy. In addition, all compounds were characterized for purity and homogeneity by high-resolution mass spectrometry (HRMS). The obtained M/z values of the compounds correspond to the calculated M/z values. According to HPLC, all the obtained products are individual compounds. All spectral data are presented in full in the Appendix A.

### 3.2. Cytokinin Activity

The CK activity of all synthesized compounds was studied in two plant bioassay systems. First bioassay was based on seedlings of a model plant *Arabidopsis thaliana* [17]. To study the effects of CK nucleosides (as CK precursors) on different CK receptors in this assay we used double insertion mutants, in which only one isoform of the CK receptors (AHK2, AHK3 or CRE1/AHK4/WOL) was active in each mutant clone, as well as the wild type (WT) plants with all three functioning receptors (Table 1).

All plants were stably transformed with the *GUS* gene fused to CK-dependent promoter of the *ARR5* gene. The hormonal activity of the compounds was determined by the level of GUS activity reflecting the intensity of the *P_ARR5_:GUS* expression [17]. Based on the results of the *Arabidopsis* bioassay (Table 1), all tested compounds were subdivided into three conventional groups, of low, medium, and high activity. The first group included compounds whose CK activity did not exceed 30% (colored blue), the second group included derivatives with medium activity from 30% to 80% (colored black), and the third highly active group exhibited activity above 80% (colored red) of the BA activity.

As an additional bioassay we used *Amaranthus* seedlings which quickly responded to CK by accumulation of the red pigment amaranthin in the dark (Table 2) [19]. This assay is rather specific and fast and considered to be a classical cytokinin bioassay [20]. As in the case of *Arabidopsis* bioassay compounds with activity values of less than 30% of BA activity were considered to be low active or inactive. Compounds with activity no less than 80% of BA activity were considered to be highly active.

### 3.3. Analysis of Results with Arabidopsis Bioassay

Ribosides of natural CKs **4**, **5**, and **7** exhibited CK effect in all experimental variants of *Arabidopsis* bioassay. Compounds **4**, **5**, and **7** exhibited the highest activity at a concentration of 10^−5^ M for the WT, leading to nearly 100% activation of CK signaling system. 10-Fold decrease in CK concentration led to approx. twofold decrease in GUS activity activation. On the other hand, compounds **4** and **5** were more active at the more physiological concentration of 10^−6^ M for the mutant clones (nearly 100% receptor activation for AHK2, 80% for AHK3 and approximately 50% for AHK4) [22]. Compound **7,** a kinetin derivative, was more active at 10^−5^ M for AHK2 and AHK4 and manifested comparable activity at both 10^−5^ M and 10^−6^ M for AHK3. *N*^6^-Phenylethyladenosine (**6**), an artificial CK derivative, demonstrated some, though weak activity at a concentration of 10^−6^ M only with the AHK2 receptor (Table 1). Two other receptors were inactive or very weakly active with **6** at 10^−6^ M.

Generally, as regards double mutant clones, our *Arabidopsis* assay makes it possible to perform a unique task, i.e. to analyze the response to CKs of individual CK receptors. In our study, the clones reacted quite uniformly, there were no case when any of CK derivatives exerted high effect with one receptor and simultaneously weak effect with another one. If WT plants or any of mutant clones demonstrated high CK activity (red color) with a defined compound, all other mutant clones showed with this compound the same (red) or at least medium (black) activity. Conversely, if WT plants or any of mutant clones demonstrated low/no CK activity (blue color) with some other compound, all other mutant clones with different receptors showed the same (blue) or medium (black) but never high (red) activity (Table 1). This may be a consequence of the use of the limited set of different CKs. In any case, such regularity can serve as an argument for the validity of the assay results.

2′-Deoxyribo-derivative of iP (**9**) was active at higher concentrations (10^−5^ M) with all individual receptors and manifested weaker activity at more physiological concentration. 2′-Deoxyribo-derivatives of aromatic CKs (**8**, **10**) were virtually inactive even at a concentration of 10^−5^ M except for derivative **11** which showed some activity at both concentrations, but only with AHK2.

5′-Deoxyribo-derivative of iP (**13**) manifested high activity for WT, medium activity for AHK2 and AHK4 and weak activity for AHK3 at 10^−5^ M and manifested weak activity for all individual receptors at 10^−6^ M. On the other hand, 5′-deoxyribo-derivatives of aromatic CKs (**12**, **14**, **15**) were inactive even at a concentration of 10^−5^ M.

Taking into account these results, several inferences can be made. 5′-Deoxyribo-derivative of iP (**13**) cannot be phosphorylated enzymatically *in vivo*, thereby the LOG-mediated one-step formation of iP is not possible here. However, since this compound exerted the CK effect, the only way for this derivative conversion into active CK is the cleavage by adenosine nucleosidase (Figure 3A). In turn, the 5′-deoxyribo-derivatives of all aromatic CKs were inactive, and since the pathway associated with the action of LOG enzyme is blocked, this LOG-dependent mechanism may be suggested to play a major role in the formation of aromatic CKs (Figure 3B). For the iP nucleoside derivatives, there were no such restrictions as deoxyribo-derivatives may be able to turn *in planta* into free bases, at least at 10^−5^ M. Therefore, the results showed that the conversion of 5′-monophosphates into active CKs via LOG-mediated cleavage in *Arabidopsis* strongly depends on the nature of the side group (aliphatic or aromatic) of the CK ribosides.

### 3.4. Analysis of Results with Amaranthus Bioassay

In the case of *Amaranthus* bioassay, *N*^6^-benzyladenosine (**4**) and *N*^6^-isopentenyladenosine (**5**) showed pronounced CK activity, while *N*^6^-phenylethyladenosine (**6**) demonstrated weak activity at 10^−6^ M (Table 2). Contrary to *Arabidopsis*, in *Amaranthus* there were no advantage of **5** over **4** in CK activity. Besides, another aromatic CK (ribo)nucleoside *N*^6^-furfuryladenosine (**7**) also showed weak activity in contrast to its medium activity in *Arabidopsis* bioassay. It is interesting that all 5′-deoxyribo-derivatives (**12**–**15**) as well as 2′-deoxyribo-derivatives (**8**–**11**) of both aromatic and isoprenoid CKs turned out to be essentially inactive in the *Amaranthus* bioassay. This peculiarity is indicative of a species specificity in the final step of active CK biosynthesis in plants.

It is known that adenosine nucleosidase, isolated from various sources, has a broad substrate specificity. Substrates for this enzyme may be adenosine, guanosine, inosine, xanthosine [23], *N*^6^-benzyladenosine [24], *N*^6^-isopentenyladenosine [25], as well as 5′-deoxyadenosine, 2′-deoxyadenosine [24] and several other purine nucleosides. Despite this generally accepted view, it cannot be excluded that the lack of activity of the deoxyribo-derivatives of aromatic CKs may be related to the substrate specificity of adenosine nucleosidase in this plant species. Another possible pathway for the formation of active CKs from 2′-deoxyribo-derivatives associated with the enzymatic phosphorylation followed by cleavage of 5′-monophosphates by the action of LOG also turned out to be unrealized. In that case, it may be due to the lack of 5′-phosphorylation of 2′-deoxyribo-derivatives by adenosine kinase, or due to the inability of cleavage of the corresponding 2′-deoxyribo-5′-monophosphates by LOG.

## 4. Conclusions

A series of nucleoside derivatives of isoprenoid CK *N*^6^-isopentenyladenine and various aromatic CKs was synthesized starting from acyl-protected ribofuranosyl-, 2′-deoxyribofuranosyl- and 5′-deoxyribofuranosyladenine derivatives using mild and efficient stereoselective *N*^6^-alkylation with further acyl deblocking, and their hormonal (cytokinin) activity was determined in two bioassays based on seedlings of *Arabidopsis thaliana* and *Amaranthus caudatus*. We demonstrated that ribo-, 2′-deoxyribo-, 5′-deoxyribo-derivatives of *N*^6^-isopentenyladenine, as well as ribosides of aromatic CKs were able to transform into active CKs and exhibited hormonal activity in *Arabidopsis*, while 5′-deoxyribo- and 2′-deoxyribo-derivatives of aromatic CKs were inactive. Therefore, not only ribo- but also 5′-deoxyribo- and 2′-deoxyribonucleosides may serve as precursors of active CKs, at least in *Arabidopsis*. However, in the latter case, the only way of biosynthesis of aromatic CKs seems to be the direct cleavage of 5′-monophosphates to active nucleobases catalyzed by phosphoribohydrolase LOG, whereas the biosynthesis of *N*^6^-isopentenyladenine may evidently proceed also by hydrolysis of CK nucleosides catalyzed by adenosine nucleosidase. As a result, it can be concluded that the biosynthesis pathways of isoprenoid (exemplified by *N*^6^-isopentenyladenine) and aromatic CKs may be different, at least in *Arabidopsis*, and that further research is needed to shed light on the mechanisms of biosynthesis of this class of compounds. Of course, at present it cannot be excluded that in other plant species the CK biosynthesis pathways are different. For example, in *Amaranthus* deoxyribo-derivatives of both *N*^6^-isopentenyladenine and aromatic CKs were equally low active. Taking into consideration the variety of biological effects of CKs on the growth, development, resistance and productivity of plants, the knowledge gained may contribute to the development of new approaches to handling the growth and development of plant species valuable to humans.

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
