# Peer review of "Distinct Peculiarities of In Planta Synthesis of Isoprenoid and Aromatic Cytokinins"

_biomolecules, 2020, doi:10.3390/biom10010086_

Round 1

Reviewer 1 Report

              The manuscript describes the preparation of a series of cytokinin ribonucleosides, 2’-deoxyribonucleosides, and 5-deoxyribonucleosides possessing either an isoprenoid or aromatic group. These compounds were then screened using Arabidopsis thaliana and Amaranthus caudatus for cytokinin activity, resulting only upon cleavage of the sugar moieties.

              The “Introduction” and “Materials and Methods” sections of the manuscript are well written and relatively thorough. The remainder of the manuscript, however, would likely benefit from a bit of reorganization and additional explanation/clarification. The “Results” section does not effectively report the data obtained. Section 3.2 is essentially a re-statement of section 2.2, which presents the methods utilized for the plant-based assays. In addition, Tables 1 and 2 a bit cumbersome and hard to read based on the spacing of the data. It would be nice to see the data presented in such a way that comparisons of the numbers could be more readily made. This should be able to be accomplished based on the 3 column layout with the ribo-, 2-deoxyribo-, and 5-deoxyribo-nucleoside general structures shown at the top and the side chain for each row shown on the left. The “Discussion” section does provide some useful descriptions of the data, but fails to provide enough information for readers to understand all of the conclusions as the authors do not put the data into context. It would be helpful to include a discussion of what is considered active or not. In addition, the significance of the various mutants utilized was not discussed. Although meaningful conclusions appear to be able to be drawn from the data, the discussion supporting these conclusions is not clear. For example, in this study only one isoprenoid CK is studied, but generalizations are made for all isoprenoids. It is not clear if this would be the case since the data for the aromatic CK’s seemed to also vary.

              Additional minor comments include: 1) the term “biotest” should likely be replaced with “bioassay” throughout the manuscript. 2) the words “convertion” in lines 75 and 76 should be “conversion”. 3) “1H-NMR-spectra” in line 98 should not be hyphenated. 4) the A in acetyl should not be capitalized in lines 120 and 121 and throughout other similar procedures. 5) In lines 123 and 124, the phrase “the reaction mixture was evaporated in vacuum” should be “the reaction mixture was concentrated in vacuo” or alternatively “under vacuum”. This also applies to subsequent procedures. 6) Line 129 and throughout the remainder of the paper, please clarify which “PrNH2” is used – isopropyl or n-propyl. 7) Line 130 and in other procedures, the phrase “was applied to column chromatography” should be “was purified via column chromatography”. 8) The sentence on lines 291 and 292 should likely be moved to the acknowledgements section. 9) Line 351, the “p” at the beginning of phenethyl should be capitalized as the first letter of the sentence. 10) the “i or ii, iii” over the arrow in scheme 1 may benefit from being “i or ii; iii” or “1. i or ii, 2. iii” to show a clear separation of steps in a sequence.

Author Response

Reviewer #1

Dear reviewer, thank you for your review of the paper. We agree with your remarks. We have carefully read the text, revised it and tried to make corrections according to your suggestions. We hope that with all the improvements the paper will be suitable for the publication.

Response to reviewer:

1) The “Results” section does not effectively report the data obtained. Section 3.2 is essentially a re-statement of section 2.2, which presents the methods utilized for the plant-based assays.

Answer: The “Results” section was extended and combined with the "Discussion" section. The added corrections are marked blue.

2) Tables 1 and 2 a bit cumbersome and hard to read based on the spacing of the data. It would be nice to see the data presented in such a way that comparisons of the numbers could be more readily made. This should be able to be accomplished based on the 3 column layout with the ribo-, 2-deoxyribo-, and 5-deoxyribo-nucleoside general structures shown at the top and the side chain for each row shown on the left.

Answer: The structures of tables 1 and 2 were significantly changed according to reviewer suggestions. Instead of full chemical structures we show now structures of the side chain at N6 position.

3) The “Discussion” section does provide some useful descriptions of the data, but fails to provide enough information for readers to understand all of the conclusions as the authors do not put the data into context. It would be helpful to include a discussion of what is considered active or not.

Answer: We made corresponding corrections in the text and marked them blue.

4) The significance of the various mutants utilized was not discussed.

Answer: The paragraph discussing the use of various Arabidopsis mutants is included, lines 386-396.

5) Although meaningful conclusions appear to be able to be drawn from the data, the discussion supporting these conclusions is not clear. For example, in this study only one isoprenoid CK is studied, but generalizations are made for all isoprenoids.

Answer: Indeed, we have studied only one isoprenoid CK (iP) and its derivative. The main cause was the complexity of chemical synthesis 5’-deoxyribo-derivatives of zeatin. In our experiments, different aromatic CKs acted rather uniformly. This might be considered as indirect evidence for the similarity of activity of isoprenoid CKs as well. However, we agree that at present it is too early to draw generalized conclusions, so we made special notes in the text that only iP derivatives were used so far.

Additional minor comments:

1) The term “biotest” should likely be replaced with “bioassay” throughout the manuscript

Answer: Corrected.

2) The words “convertion” in lines 75 and 76 should be “conversion”.

Answer: Corrected.

3) “1H-NMR-spectra” in line 98 should not be hyphenated.

Answer: Corrected.

4) The A in acetyl should not be capitalized in lines 120 and 121 and throughout other similar procedures.

Answer: Corrected. The same corrections were made in lines 228 and 229.

5) In lines 123 and 124, the phrase “the reaction mixture was evaporated in vacuum” should be “the reaction mixture was concentrated in vacuo” or alternatively “under vacuum”. This also applies to subsequent procedures.

Answer: The sentences in both procedures were rephrased to “concentrated in vacuo”

6) Line 129 and throughout the remainder of the paper, please clarify which “PrNH2” is used – isopropyl or n-propyl.

Answer: In all our procedures the deacetylation step was carried out in the presence of n-PrNH2. The corresponding corrections were made.

7) Line 130 and in other procedures, the phrase “was applied to column chromatography” should be “was purified via column chromatography”.

Answer: These sentences were rephrased to: “…the residue was purified by column chromatography on silica gel.”

8) The sentence on lines 291 and 292 should likely be moved to the acknowledgements section.

Answer: Corrected.

9) Line 351, the “p” at the beginning of phenethyl should be capitalized as the first letter of the sentence.

Answer: Corrected.

10) the “i or ii, iii” over the arrow in scheme 1 may benefit from being “i or ii; iii” or “1. i or ii, 2. iii” to show a clear separation of steps in a sequence.

Answer: Corrected.

Reviewer 2 Report

Dear Authors,

1) Please transport all the NMR text of the synthesized compounds into the supplementary document. In addition, please remove all the redundant information such as multiplicity, coupling constants, proton and carbon positions from the NMR text, leaving only with 1H and 13C chemical shifts.

2) Please improve the sentences from abstract to main text to conclusion in the manuscript as many of these were exactly same to below publications [i,ii].

(i) Comparative analysis of the biosynthesis of isoprenoid and aromatic cytokinins, 2019, Doklady Akademii nauk / [RossiÄ­skaia akademii nauk] 488(6):673-676. [DOI: 10.31857/S0869-56524886673-676]

(ii) Comparative analysis of the biosynthesis of isoprenoid and aromatic cytokinins, 2019. Doklady Biochemistry and Biophysics 488(6):346-349. [DOI: 10.1134/S1607672919050156]

3) With similar reason as above, please improve the Figures 1 and 2.

4) Please remove all the structures from the Tables 1 and 2, arranged table into a simple form, where x-axis of the table will have four columns which are WT, AHK2, AHK3, AHK4, then y-axis of the table will have two rows for each compound  (10-5 M and 10-6M). 

5) The reported activities of compounds 4, 5, 6, 12, 13 and 14 from the above publications [i,ii], can be included into Tables 1 and 2 for the sake of ease comparing the activity by readers, however Authors must clearly indicate these compounds were previously reported together with the references in the Table, and also mention in the main text with references. 

Author Response

Reviewer #2

Dear reviewer, thank you for your review of the paper. We agree with your remarks. We have carefully read the text, revised it and tried to make corrections according to your suggestions. We hope that with all the improvements the paper will be suitable for the publication.

Response to reviewer:

1) Please transport all the NMR text of the synthesized compounds into the supplementary document. In addition, please remove all the redundant information such as multiplicity, coupling constants, proton and carbon positions from the NMR text, leaving only with 1H and 13C chemical shifts.

Answer: We cannot agree with this comment, since all NMR data, including chemical shifts and spin-spin coupling constants (J), are important characteristics of the obtained compounds. These characteristics are usually standard in the description of chemical compounds and may be useful to readers. Therefore, we decided to leave this issue to the discretion of the editor.

2) Please improve the sentences from abstract to main text to conclusion in the manuscript as many of these were exactly same to below publications [i,ii].

(i) Comparative analysis of the biosynthesis of isoprenoid and aromatic cytokinins, 2019, Doklady Akademii nauk / [RossiÄ­skaia akademii nauk] 488(6):673-676. [DOI: 10.31857/S0869-56524886673-676]

(ii) Comparative analysis of the biosynthesis of isoprenoid and aromatic cytokinins, 2019. Doklady Biochemistry and Biophysics 488(6):346-349. [DOI: 10.1134/S1607672919050156]

Answer: The text of the manuscript was improved and big textual identities were removed. Added corrections are marked blue. The corresponding reference on our preliminary communication in “Dokl. Biochem. Biophys” (translated English version of Russian-language “Dokl. Akademii nauk”) was provided. This journal publishes a preliminary data in the form of short communications that allow publishing the full investigations in other journals.

3) With similar reason as above, please improve the Figures 1 and 2.

Answer: The figures 1 and 2 were changed.

4) Please remove all the structures from the Tables 1 and 2, arranged table into a simple form, where x-axis of the table will have four columns which are WT, AHK2, AHK3, AHK4, then y-axis of the table will have two rows for each compound  (10-5 M and 10-6M).

Answer: The structures in tables 1 and 2 were significantly changed. Instead of full chemical structures, we show structure of side chains at N6 position (according to the reviewer comment).

5) The reported activities of compounds 4561213 and 14 from the above publications [i,ii], can be included into Tables 1 and 2 for the sake of ease comparing the activity by readers, however Authors must clearly indicate these compounds were previously reported together with the references in the Table, and also mention in the main text with references. 

Answer: The preliminary biological data for some of compounds (4-6 and 12-14) were published earlier as short communication, the respective reference was provided. Table 1 (Arabidopsis bioassay) includes unpublished data partly, Table 2 (Amaranthus caudatus bioassay) includes only unpublished data.

Round 2

Reviewer 2 Report

Dear Authors,

Thank you for addressing the points.

It is not necessary to change the Figures 1 and 2 as improvement. If Authors favor the previous figures in this manuscript, Authors might need to obtain the copyright from Author's previous journals.